# Postural Orthostatic Tachycardia Syndrome (POTS) as an Adverse Event to the Human Papilloma Virus (HPV) Vaccine and Its Relationship with Ehlers–Danlos Syndrome (EDS)

**DOI:** 10.3390/reports7020036

**Published:** 2024-05-12

**Authors:** Nicole Schipperijn, Megan Wijesinghe, Aisa Romo, Benjamin Brooks

**Affiliations:** College of Osteopathic Medicine, Rocky Vista University, Ivins, UT 84738, USA; megan.wijesinghe@co.rvu.edu (M.W.); aisa.romo@ut.rvu.edu (A.R.); bbrooks@rvu.edu (B.B.)

**Keywords:** Ehlers–Danlos syndrome, vaccine adverse events, postural orthostatic tachycardia syndrome

## Abstract

Gardasil 4, a human papilloma virus vaccine, has been shown to protect against various cancers, including cervical cancer. Common side effects include injection site pain, fever, headaches, and muscle aches. In some individuals, the severe side effect of postural orthostatic tachycardia syndrome (POTS) has been reported. POTS is characterized by the abnormal response of lightheadedness, blurry vision, and dizziness while transitioning to an upright posture. POTS predominately affects women, with more than eighty-five (85) percent of POTS patients being female. POTS, on average, takes five years and eleven months to receive diagnosis. Additionally, a strong association between POTS and Ehlers–Danlos Syndrome Type III (EDS) exists. Eighty (80) percent of patients with EDS have POTS. This severe side effect indicates that providers need to be aware of this strong association of HPV vaccinations and POTS. In this report, we will present a case of a young women with a past medical history significant for EDS type III who was diagnosed with POTS after receiving Gardasil 4 vaccination. This case demonstrates the need for physicians to be aware of the association of POTS with EDS type III and HPV vaccination. Physician awareness of the associations, signs, and symptoms of POTS and earlier testing at the first presentation of signs and symptoms will limit the negative impact on patient’s quality of life.

## 1. Introduction

The Gardasil 4 vaccine, which protects against nine strains of the Human Papilloma Virus (HPV), was first approved in 2006. Gardasil 9, currently the most frequently administered HPV vaccine, was approved in December of 2014 [1] and has been shown to reduce the risk of developing various cancers, including cervical cancer [2]. Common side effects of the vaccine include injection site pain, fever, headaches, and muscle aches; however, the vaccine has been known to cause severe adverse events for some individuals, such as postural orthostatic tachycardia syndrome (POTS) [1].

POTS is a form of orthostatic intolerance associated with excessive tachycardia upon standing, which is typically relieved by recumbence and often accompanied by many other symptoms [3,4]. Before the COVID-19 pandemic, POTS was estimated to impact 1 to 3 million Americans, with the majority being women between the ages of 15 and 50 [5,6]. POTS morbidity derives primarily from symptom burden and functional impairment, including profound fatigue, cognitive dysfunction, headaches, gastrointestinal disturbance, syncope, and presyncope, which limit physical activity [3,4]. POTS has significant impacts on daily life, including a 52% unemployment rate among patients, 70.5% with income loss, and 95% with significant medical expenses. POTS-related work and financial instability highlight the need for improved diagnoses and treatments [7].

POTS can be triggered by diverse initiating events, often making its exact initiating event challenging to pinpoint. Commonly, POTS is preceded by a viral infection, which can disrupt the autonomic nervous system’s functioning. In some individuals, POTS may also emerge following significant physical or emotional stress, surgery, or pregnancy, indicating a potential link to bodily stressors and changes [8]. Additionally, while rare and not well understood, there are documented cases where POTS symptoms have developed after vaccination [9,10]. While the initiating events are poorly understood, some potential risk factors have been identified. Further, evidence is limited for vaccination, including the HPV vaccination, in the initiating events for POTS [11].

Ehlers–Danlos Syndrome (EDS) Type III, now commonly referred to as Hypermobile EDS (hEDS), is considered by some to be a risk factor for Postural Orthostatic Tachycardia Syndrome (POTS). The joint hypermobility and connective tissue fragility characteristic of hEDS can impact the cardiovascular system, potentially predisposing individuals to POTS [12]. This relationship highlights the potential for shared underlying pathophysiological mechanisms between hEDS and POTS, possibly leading to common symptoms such as rapid heart rate upon standing and autonomic dysregulation [13].

Here, we report on one young woman with a past medical history of hEDS who was diagnosed with POTS after receiving the Gardasil 4 vaccine. This case highlights the importance of physicians’ ability to recognize the associations, signs, and symptoms of POTS to reach a quicker diagnosis and highlights associated with EDS, POTS, and HPV vaccination.

## 2. Case Presentation Section

A 42-year-old female received the Gardasil 4 vaccine on 14 January 2014. Her past medical history indicated migraines and EDS type III. The patient was diagnosed with EDS in 2010 and presented with a Beighton score of 7 out of 9. Medications the patient was taking at time of vaccination include bupropion 100 mg twice daily, fluoxetine 60 mg once daily, indomethacin 25 mg three times daily, and topiramate 100 mg twice daily.

A few days later, on 20 January 2014, the patient returned to the clinic complaining of unilateral left eye pain, a severe headache, and experiencing symptoms of POTS, specifically tachycardia. The patient experienced frequent bouts of syncope, impacting her daily life. The patient stated that when her POTS worsened, she had more than 20 POTS episodes per day. While the patient began experiencing symptoms of POTS in 2014, it took several doctor visits complaining of POTS symptoms before a table tilt test was performed. This resulted in a diagnosis not being made until 2021. Table 1 shows the patient’s positive table tilt test with pulses measuring to a maximum of 158 bpm with seventy (70) degrees of tilting. After the patient’s POTS diagnosis, Modafinil was prescribed and resolved the patient’s POTS symptoms. 

Adverse symptoms of POTS following the Gardasil 4 vaccine has had a severe impact on the patient’s daily life. The patient claimed she went from a very sociable person to an introvert. The patient fears having a POTS episode when she is alone. She also fears leaving home and having an episode away from home.

## 3. Discussion

Case reports have documented adolescent girls developing symptoms of orthostatic intolerance and excessive tachycardia consistent with POTS in the days and weeks following Gardasil immunization [10,11]. Butts et al. provide a review of the current literature on the potential association between the HPV vaccine and POTS [12]. Chandler et al. discuss safety concerns with the HPV vaccine and reports of adverse events, including POTS [1,13]. Barboi et al. (2020) present a position statement from the American Autonomic Society on the HPV vaccine and autonomic disorders, including POTS [14]. The mechanisms linking the Gardasil vaccination to POTS are not well understood but may involve autoimmune or autonomic dysfunction triggered by the vaccine. Further research is warranted to elucidate the potential association between POTS and the Gardasil vaccination, given the potential impact on the lives of young vaccine recipients.

There is evidence that HPV vaccinations can trigger mast cell activation syndromes [15]. This can explain the association of HPV-triggered POTS and the association of EDS type III. Mast cell syndrome, specifically mast cell activation syndrome (MCAS), is a disorder characterized by the abnormal release of mast cell mediators, leading to a wide variety of symptoms, including itching, flushing, abdominal pain, and anaphylaxis. Mast cells are key players in allergic reactions and immune responses, but in MCAS, they become hyper-reactive, releasing their contents in response to triggers that would not normally provoke a reaction. Research indicates that MCAS can be triggered by various factors, including infections, stress, and environmental exposures. MCAS can present with diverse symptoms affecting multiple organ systems, often leading to misdiagnosis and undertreatment [16]. However, MCAS also has a distinct clinical treatment because it may respond to targeted therapy [11]. Up to 30 percent of POTS patients have mast cell activation syndrome, and it is well known that EDS type III is a mast cell activation disorder [17,18,19]. POTS as an adverse event to HPV vaccine and the connection with EDS could be explained by mast cell activation causing POTS and EDS to present in patients post HPV-vaccination.

POTS symptoms can take months or even years to manifest fully, making accurate tracking and quantification of POTS incidence following vaccination challenging. Furthermore, the lack of awareness of POTS diagnostic criteria among providers and terminology varying from postural orthostatic tachycardia syndrome to chronic fatigue syndrome confound efforts to definitively diagnose and document POTS cases that may be linked to vaccinations. These factors make determining the true prevalence and causal association between POTS and vaccination, like Gardasil, difficult to ascertain [20,21,22,23].

POTS can have a significant impact on patients’ lives. Some patients report mild symptoms and can continue normal daily activity. Others live through symptoms severe enough that everyday activities, such as walking, bathing, housework, and even eating, are significantly limited [3,4]. High out-of-pocket medical costs related to POTS are a burden on patients seeking a diagnosis or treatment [7].

Accurate diagnosis of POTS is a crucial yet difficult milestone for patients. POTS can be misdiagnosed because patients may present with a variety of symptoms without any clinically significant findings. Physicians may misdiagnose POTS as anxiety, panic attacks, vasovagal syncope, or inappropriate sinus tachycardia. Many patients are inappropriately diagnosed with psychiatric disorders and consequently may distrust the medical community [24,25,26]. Symptoms can be present for months or years before a final diagnosis of POTS is made. Some patients may wait two to seven years before receiving a diagnosis, and many patients who first suggest a diagnosis of POTS to their physicians must present symptoms to several practitioners before receiving a POTS diagnosis [25]. In this case, it took six years before a table tilt was performed and POTS was diagnosed. Additionally in our patient’s case, there were minimal diagnostic tests run to determine cardiac and hemodynamic status, adding to the time to POTS diagnosis and further showing clinicians’ lack of awareness of POTS symptoms.

The long period from symptom presentation to diagnosis is frustrating for patients because medical costs can accumulate, and they may not receive the proper medical treatment when the correct diagnosis is not made. Knowing the strong association between hEDS and POTS, and POTS as a potential adverse event to the HPV vaccine, is crucial to reduce the time between the presentation and diagnosis of POTS. Patients with hEDS should be more frequently tested for POTS, especially after receiving an HPV vaccine. Patients with hEDS should also be educated about the potential risk for developing POTS as part of pre-vaccination education.

This report is not meant to be construed as an anti-vaccination statement but rather as an opportunity to improve patient care.

A potential association between POTS and hEDS exists [24]. We recognize that the incidence of POTS associated with hEDS is low; however, given the clear considerations and impacts for patients with hEDS, additional monitoring in these cases is warranted before administering HPV vaccinations, as the vaccine may initiate POTS. Here, we recommend additional research and vigilance to identify potential associations with patients with hEDS developing POTS after HPV vaccination.

## Figures and Tables

**Table 1 reports-07-00036-t001:** Vital signs recorded during the patient’s table tilt test. Results show a positive table tilt test diagnostic for POTS.

Degree of Tilt	17 February 2021 1020	17 February 2021 1019	17 February 2021 1018
	70 Degrees	70 Degrees	70 Degrees
Heart Rate	144 bpm	145 bpm	136 bpm
Blood Pressure	115/82	122/90	132/84

## Data Availability

Supporting data may be found in the Vaccine Adverse Event Reporting System.

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
