# Peer review of "Postural Orthostatic Tachycardia Syndrome (POTS) as an Adverse Event to the Human Papilloma Virus (HPV) Vaccine and Its Relationship with Ehlers–Danlos Syndrome (EDS)"

_reports, 2024, doi:10.3390/reports7020036_

Round 1

Reviewer 1 Report

Comments and Suggestions for Authors

I was invited to revise the paper entitled "Postural Orthostatic Tachycardia Syndrome (POTS) as an Adverse Event to the Human Papilloma Virus (HPV) Vaccine and its Relationship to Ehlers Danlos Syndrome (EDS)". It was a case report reporting the developing of POTS after the administration of Gardasil 9 vaccination.

Major observations:

- Lines 34-35 are not supported by adequate references;

- In abstract and in line 67 Authors cited Gardasil 4 vaccine, meanwhile in introduction section they described Gardasil 9 vaccine;

- Authors did not reported the texact timing (date) of the vaccination and its association with the first POTS episode.  It is well known that the onset of POTS may be precipitated by a typical immunological stressor such as viral syndrome (often upper respiratory or gastrointestinal) or physical trauma (such as concussion) (ref

Vernino S, Bourne KM, Stiles LE, Grubb BP, Fedorowski A, Stewart JM, Arnold AC, Pace LA, Axelsson J, Boris JR, Moak JP, Goodman BP, Chémali KR, Chung TH, Goldstein DS, Diedrich A, Miglis MG, Cortez MM, Miller AJ, Freeman R, Biaggioni I, Rowe PC, Sheldon RS, Shibao CA, Systrom DM, Cook GA, Doherty TA, Abdallah HI, Darbari A, Raj SR. Postural orthostatic tachycardia syndrome (POTS): State of the science and clinical care from a 2019 National Institutes of Health Expert Consensus Meeting - Part 1. Auton Neurosci. 2021 Nov;235:102828. doi: 10.1016/j.autneu.2021.102828). So, without an exact report of jab intake and the first clinical episode, Authors are not able to speculate about the association; - How stated by Authors, an association between POTS and HD is knonw and frequently described, so this report conclusion were biased. It is clear that vaccination is only a confounder.

Reviewer 2 Report

Comments and Suggestions for Authors

The authors submitted the case report in which they tackled the POTS with EDS after vaccination gegen HPV. Overall, the findings seem to be impressive, while there are several concerns about the interpretation of them.

1. The authors indicated that POST is a risk factors for POTS in EDS patients. However, the patient' description including the status, condition and other parameters did not present clarity of the causes and consequences of the complication. Please, extend the section Methods and Results so that more information of the patient would be added including its status, hemodynemics, cardiac structure and fuction, lab data occures .

2. Please, add  short description in connection with suggestion of what causes / reasons are likely to be made out to explain the indings (cardiomyopathy including ion channel alterations, etc)

3. Concomitant medication should reported and discussed

4. Examples of ECGs, Echo-gramms, other visualisation procedures are welcome

Reviewer 3 Report

Comments and Suggestions for Authors

Dear Authors,

This clinical case is very interesting because it reveals adverse effects of a vaccine.

From a technical point of view I think it is a very good job.

Due to my training, I have no further comments to make.

Author Response

We appreciate the thorough review of our manuscript and your kind words. Thank you for your time. 

Reviewer 4 Report

Comments and Suggestions for Authors

Dear authors,

Your case report, Postural Orthostatic Tachycardia Syndrome (POTS) as an Adverse Event to the Human Papilloma Virus (HPV) Vaccine and  its Relationship to Ehlers Danlos Syndrome (EDS) is interesting and useful to be presented.

I believe case reports are important because they show various experience.

HPV vaccination is useful and important and in some countries is not yet completed.

So, your case report came in handy.

The introduction provides sufficient background and include all relevant references. The references cited are good for the manuscript.

The design of the case report presentation is fine. Discission supports the findings of the research.

I recommand publication as it is.

Author Response

(The authors gave the same response as above.)

Round 2

Reviewer 2 Report

Comments and Suggestions for Authors

The authors submitted the revised version of the paper along with thoroughly prepared reply to reviewers. I have no serious concerns about the article in its revised version.